# Experiences of 'traditional' and 'one-stop' MRI-based prostate cancer diagnostic pathways in England: a qualitative study with patients and GPs

Samuel William David Merriel [iD],[1] Stephanie Archer [iD],[2,3] Alice S Forster [iD],[4] David Eldred-Evans,[5] John McGrath,[6] Hashim Uddin Ahmed [iD],[5] Willie Hamilton [iD],[1] Fiona M Walter [iD][2,7]

For numbered affiliations see end of article.

**Correspondence to**
Samuel William David Merriel;
s.w.d.merriel@exeter.ac.uk

## ABSTRACT

**Objectives** This study aimed to understand and explore patient and general practitioner (GP) experiences of 'traditional' and 'one-stop' prostate cancer diagnostic pathways in England.

**Design** Qualitative study using semi-structured interviews, analysed using inductive thematic analysis

**Setting** Patients were recruited from National Health Service (NHS) Trusts in London and in Devon; GPs were recruited via National Institute for Health Research (NIHR) Clinical Research Networks. Interviews were conducted in person or via telephone.

**Participants** Patients who had undergone a MRI scan of the prostate as part of their diagnostic work-up for possible prostate cancer, and GPs who had referred at least one patient for possible prostate cancer in the preceding 12 months.

**Results** 22 patients (aged 47–80 years) and 10 GPs (6 female, aged 38–58 years) were interviewed. Patients described three key themes: *cancer beliefs* in relation to patient's attitudes towards prostate cancer; *communication* with their GP and specialist having a significant impact on experience of the pathway and *pathway experience* being influenced by appointment and test burden. GP interview themes included: the challenges of dealing with *imperfect information* in the current pathway; *managing uncertainty* in identifying patients with possible prostate cancer and sharing this uncertainty with them, and other social, cultural and personal *contextual influences*.

**Conclusions** Patients and GPs reported a range of experiences and views of the current prostate cancer diagnostic pathways in England. Patients valued 'one-stop' pathways integrating prostate MRI and diagnostic consultations with specialists over the more traditional approach of several hospital appointments. GPs remain uncertain how best to identify patients needing referral for urgent prostate cancer testing due to the lack of accurate triage and risk assessment strategies.

## INTRODUCTION

Patient experience of healthcare has developed as an important marker of quality of care in recent decades. However, measuring

---

**STRENGTHS AND LIMITATIONS OF THIS STUDY**

⇒ Patient experiences of two very different prostate cancer diagnostic pathways compared and contrasted.

⇒ Patient sample feature a broad range of ages, geographical regions and cancer investigation journeys to generate rich data.

⇒ First study to explore general practitioner (GP) experience and understanding of new prostate cancer diagnostic pathways incorporating MRI.

⇒ Limited knowledge of prostate MRI curtailed interviews with some GP participants.

---

and understanding patient experience of diagnostic pathways and services is underexplored and poorly prioritised compared with other aspects of healthcare such as access or treatments.[1] Assessment of the impact of variations in pathway design between health services may also identify elements associated with better patient experience, such as quicker access to testing, that could be implemented more widely and elements associated with adverse experience, such as high appointment burden, that can be avoided.

The National Health Service (NHS) in England has a Two Week Wait (2WW) urgent cancer referral pathway system, where any patient with symptoms or signs of a potential undiagnosed cancer referred by their general practitioner (GP) should have a specialist review for further investigation within 2 weeks.[2] Cancer diagnostic pathways are prioritised for urgent access to specialist assessment and diagnostic tests as early-stage diagnosis is associated with increased survival.[3] Not only do shorter diagnostic intervals improve outcomes for patients, but patients also report better experiences of care.[4] Significant variation in cancer

diagnostic pathways between NHS Trusts and regions in England exists, most markedly for prostate cancer.[5] Identifying patients for 2WW prostate cancer referral in primary care is also challenging for GPs owing to limitations of existing tests, including prostate-specific antigen (PSA), which can impact on doctor–patient communication and patient experience of the early stages of the prostate cancer diagnostic pathway.[6 7]

National Institute for Health and Care Excellence (NICE) guidance for diagnosing prostate cancer in England was updated in 2019 to recommend prebiopsy MRI for men suspected of having prostate cancer.[8] In response, Cancer Alliances and Hospital Trusts in the NHS have updated local prostate cancer diagnostic pathways, with significant variation in the implementation of MRI.[9] Despite the potential benefits, prostate MRI brings in terms of more accurate prostate cancer diagnosis,[10] adding further testing into the prostate cancer diagnostic pathway could lengthen the diagnostic interval, adversely impacting patient experience. Experiences of the prostate cancer diagnostic pathway for patients and GPs since the advent of prostate MRI is unknown. The aim of this study was to elicit the experience of patients and GPs following two prostate cancer diagnostic pathways that incorporate prebiopsy MRI in different ways to inform optimal prostate cancer diagnostic pathway design. In the context of the Model of Pathways to Treatment, a key theoretical framework in cancer diagnostic pathways, this study focuses on the 'Help-seeking' and 'Diagnostic' intervals and explores the perspectives of both patient and clinician.[11 12]

## METHODS

This qualitative study used semistructured interviews to explore the experiences of patients referred from primary care with possible prostate cancer who had undergone an MRI, and GPs who have referred men with possible prostate cancer for further investigation. A constructivist approach was adopted to access the data and understand the experiences of patients and GPs[13] based on their individual experiences (past and present) and the sociocultural context.[14 15]

## Participants

This study recruited participants from two populations through purposive sampling:
► Patients with possible prostate cancer who had undergone an MRI as part of their diagnostic workup.
► GPs who had referred at least one patient for investigation for possible prostate cancer within the preceding 12 months.

Patients who were undergoing MRI for active surveillance or watchful waiting for a previously diagnosed prostate cancer were not eligible, as the focus of this study was on the role of MRI in the diagnosis of prostate cancer rather than management.

## Recruitment

Patients were recruited from two NHS Trusts in England: The Royal Devon & Exeter NHS Foundation Trust in Exeter and the Imperial College Healthcare NHS Trust in London. The Royal Devon & Exeter Hospital use separate outpatient appointments in the South West (SW) Prostate Cancer Diagnostic Pathway for a prostate MRI, consultant review and prostate biopsy (if required), as shown in figure 1. Imperial College employ the RAPID pathway, where patients undergo a prostate MRI scan, receive their MRI result and potentially undergo a prostate biopsy on the same day at a single outpatient attendance (see figure 2). These Trusts were selected as prostate MRI has been implemented in very different ways, creating the opportunity to explore and compare patient and clinician experiences in different clinical contexts. Research staff at the Trusts identified potentially eligible men and contacted them within days of undergoing an MRI to discuss this study and offer the men a Patient Information Leaflet (PIL). The lead investigator and local recruitment leads were in regular contact throughout recruitment to identify any under-represented groups of men and focus recruitment where needed. Travel costs for patient participants to attend a face-to-face interview were reimbursed, and participants were also offered a gift voucher in recognition of contributing their time.

GPs were recruited through two National Institute for Health Research (NIHR) Clinical Research Networks (CRNs) in the same regions as the hospital sites: Northwest

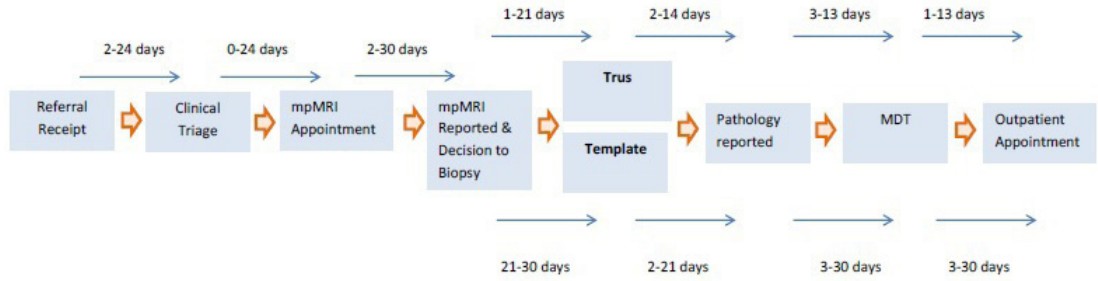

**Figure 1** South West Prostate Cancer Diagnostic Pathway, NHS Cancer Alliances in South-West Peninsula and Somerset, Wiltshire, Avon and Gloucester (SWAG). mpMRI, multiparametric MRI; TRUS, transrectal ultrasound guided biopsy; MDT, multidisciplinary team.

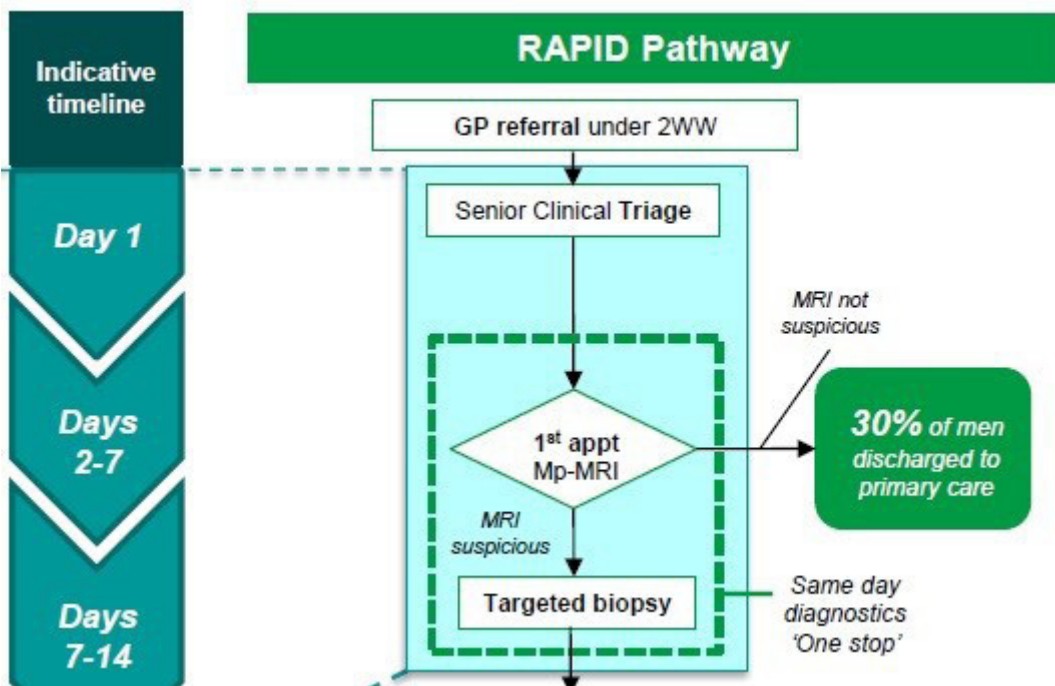

**Figure 2** RAPID pathway, Imperial College Healthcare NHS Trust, London. mpMRI, multiparametric MRI; 2WW, Two Week Wait pathway.

London CRN and the South-West Peninsula CRN. The CRNs promoted the study to local practices, and GPs expressed their interest to the CRNs. Eligibility and basic demographics were checked to assist with purposive sampling. GPs chosen for invitation into the study were given a PIL to review prior to the arrangement of an interview. GP practices were reimbursed for the GP's time to participate in the study.

A purposive sampling approach was used, in order to obtain a diverse group of participants with a wide range of geographical locations, ages, ethnicities, genders (GPs) and MRI results (patients).

### Data collection

One-to-one interviews were conducted with all participants in this study between July and November 2019 by SWDM (a male GP with training in qualitative interviewing). Patient participants were either interviewed face-to-face in their own home or via telephone, while all GP participant interviews were conducted via telephone. The interviewer and participant were not known to each other before the study. Formal written consent was obtained from all participants, and patient's partners if present (n=2), prior to commencement of the interview. A semistructured approach was followed, with separate interview topic guides for patient and GP interviews to support discussions (see online supplemental files 1 and 2). The topic guide was developed iteratively with input from our patient/public partners; it was further refined during the first three interviews to incorporate all aspects of the revised prostate cancer diagnostic pathway and was used flexibly to ensure that no key aspects of the diagnostic pathway experience were missed. An encrypted

audio recording device was employed to record all interviews, and written notes were taken during and immediately following the interviews. Interview times ranged between 15 and 45 min each, and no repeat or follow-up interviews were undertaken. Interview recordings were transferred securely to an independent transcribing service, and transcribed verbatim.

### Data analysis

An inductive thematic analysis was conducted to understand the experiences of participants,[16] using the conceptual framework of the Model of Pathways to Treatment.[11 12] The researchers initially immersed themselves in the data through reading and rereading individual transcripts and listening back to the audio recordings of the interviews. A selection of early interviews were coded, and this initial coding framework was reviewed and refined by SWDM, SA and FMW. The remaining interview transcripts were coded by SWDM inductively from the entirety of the data. The codes were reviewed and arranged into themes through an iterative process, returning to the original data as needed. Patient and GP transcripts were analysed separately. Within and between themes, the experiences of participants following different diagnostic pathways were compared and contrasted. Recruitment ceased when no new themes emerged in analysis. Transcripts were imported into NVivo V.12 to manage the data for the analysis. A study summary report was sent to all study participants after completion of data analysis.

### Patient

Eight men were recruited via the People in Health West of England initiative to contribute to the research:

**Table 1** Patient and GP demographics

| | Patients (n=22) | | GPs (n=10) | |
|---|---|---|---|---|
| Age | | | Age | |
| <65 | 8 | | 31–40 | 3 |
| 65+ | 14 | | 41–50 | 6 |
| | | | 50+ | 1 |
| Location | | | Location | |
| London | 10 | | London | 4 |
| Devon | 12 | | Devon | 6 |
| Ethnicity | | | Gender | |
| White | 19 | | Male | 4 |
| Non-white | 3 | | Female | 6 |
| PIRADS v2 | | | Role | |
| 1–2 | 6 | | Partner | 8 |
| 3–5 | 15 | | Salaried | 2 |
| Unknown | 1 | | | |

*PIRADS, Prostate Imaging-Reporting And Data System v2 score of 1–2 suggest clinically significant prostate cancer is unlikely and biopsy not indicated. A PIRADS score of 3–5 indicates at least one suspicious area of the prostate that warrants biopsy.
GP, general practitioner.

these men had a range of ages, locations, ethnic backgrounds and experiences with prostate cancer. PPI group members reviewed the plain English summary and all patient participant documents and gave feedback prior to submission as part of the ethical approval application. PPI group members also gave input into the interview topic guides and the expected burden of involvement for participants. One of the anonymised patient interview transcripts was shared with the group at a meeting and discussed to explore themes emerging from the text.

### COREQ reporting guidelines

This manuscript has been written in accordance with the consolidated criterion for reporting qualitative research (COREQ) checklist.[17] Further detail regarding the methods can be found in the study protocol (see online supplemental file 3).

### RESULTS

#### Participants

Twenty-two patients were interviewed between July and November 2019; two with their wives present and involved in the interview: participant ages ranged from 47 to 80 years. Ten GPs were interviewed: most were female (n=6), with an age range of 38–58 years (see table 1). Five further potential (three patients and two GPs) participants were approached but declined to participate without giving a reason.

#### Patient experiences of the prostate cancer diagnostic pathway

We identified three main themes with interlinking subthemes (see figure 3): cancer beliefs, communication and pathway experience.

##### Cancer beliefs

The decision for patients to see their doctor about potential prostate problems was not undertaken in isolation (*Outside influences*). The experiences of family members and friends shaped the patients' expectations, and family members and partners often encouraged men to be tested:

> Obviously back then he [dad] was in his mid to late 60s. And I think I didn't really know about it until he'd gone for his MRI and got the results and everything, and then all of a sudden he sat me down and told me all about it. P20 (London, <65)

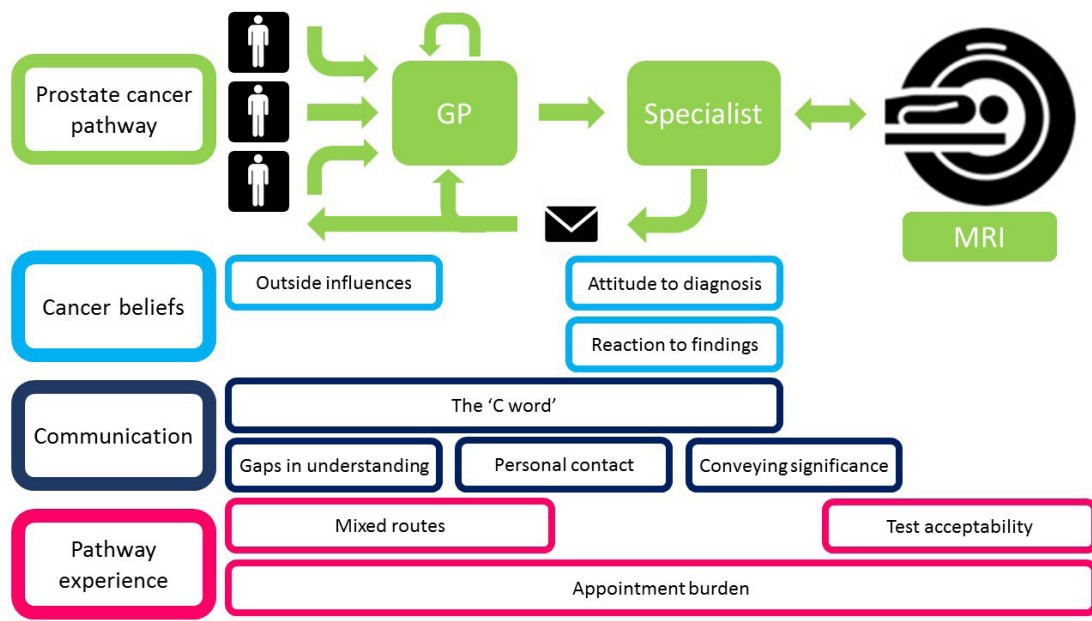

**Figure 3** Thematic diagram from patient participant interviews. GP, general practitioner.

Most patients' attitudes towards the possibility of a diagnosis of cancer (*Attitude to diagnosis*) were fairly relaxed. Many seemed philosophical about the prospect:

it is what it is P03 (Devon, <65)

The reactions of patients who had a diagnosis of prostate cancer (*Reaction to findings*) were mixed, ranging from despondence to quick acceptance:

'Not fair. No, it's… it's not fair on… on anyone, not just me. It isn't fair on anyone.' P01 (Devon, 65+)

### Communication

The absence of the use of the word cancer ('*C word*') was evident in interviews with many patients. Patients also reported a reluctance from clinicians to raise cancer specifically as a possibility during a consultation, even if they were referred for urgent tests to rule out a diagnosis of prostate cancer:

And then this developed. P01 (Devon, 65+)

For me, my… my dad had it roughly about eight, nine years… eight to ten years ago, I suppose. He had it. P20 (London, <65)

The only thing that I found was you were given leaflets that mention a lot about cancer but no one actually really, sort of like said to me, you know, there's a possibility that you could have cancer or you know, that you're just being given leaflets and such, and no one really explained to you that there is a possibility. P25 (Devon, <65)

The mode of communication to the patient from clinicians (*personal contact*) appeared to directly affect their experiences of the pathway. Most London patients sat down with their consultant and reviewed their MRI results together, whereas many patients in Devon received their results via a letter:

I think it was interesting to see this sort of slightly darker little, ti… little circular area that he thought might be cancerous and… and also explain that they would need to take some samples from another area which… which was more the normal colour of the whole gland for comparison. P13 (London, 65+)

Most of the letters go to the GP and I just get a copy." P23 (Devon, 65+)

Communicating the meaning (*conveying significance*) of the results of the MRI and other tests performed was very important to help patients and their partners understand what the results mean for them as an individual:

Yeah, so apparently, because this is mid-rank they said that if you just got the first circle, the first ones in, they probably wouldn't have done anything about it and you could have had a lot of years where you just monitor that. But because P03 was mid-stage, they said we have to do something. P03's partner (Devon, <65)

Despite most of the patients having undergone a prostate MRI by the time of their interview, there were still limited understanding of the MRI results for some patients (*Gaps in understanding*). More patients from Devon reported these gaps, which often appeared to be a result of communication breakdown between the patient and the doctor:

Umm… I think, all I know is those letters passed to and fro between the urologist and my GP, and I'm copied in on these things and there was some mention of an abnormality on the left hand side or somewhere or other on the prostate. That's all I know. P23 (Devon, 65+)

### Pathway experience

Patients entered the pathway in different ways, with varied length of time and diagnostic work-up prior to urgent suspected cancer referral (*Mixed routes*). For patients in Devon, the prostate cancer pathway required a number of individual appointments, whereas most patients in London received their MRI results on the same day or soon after which was well received (*Appointments burden*):

I had a PSA of, I think it was 4.03, which was fractionally above the four limit. Then they gave me two additional PSAs every three months, so I went back three months later did another PSA and then I think it was about 3.84. Then another one three months later was 4.08. So then I saw a urologist at Exeter and as a precaution they gave me an MRI and the MRI identified an area of concern if you like [inaudible]. Then I had a biopsy and what that identified was that the area of concern that the MRI identified, there was no cancer, but there was cancer in another area. P04 (Devon, 65+)

so… the scan, you get the result within minutes, and even though I had to wait perhaps an hour before I actually saw the doctor but that's a lot less than three months." P05 (London, 65+)

Patient interviewees were generally positive about undergoing investigations for possible prostate cancer, including blood tests and MRI. Most, but not all, patients reported that undergoing an MRI of the prostate was not a significant undertaking (*Test acceptability*):

I'd go for any scan, anything like that. Needles don't bother me, scans don't bother me." P21 (Devon, 65+)

### GP experiences of the prostate cancer diagnostic pathway

We identified three main themes: imperfect information, managing uncertainty and contextual influences (see figure 4).

### Imperfect information

GPs spoke at length about the limitations of the current primary care diagnostic pathway for prostate cancer, and

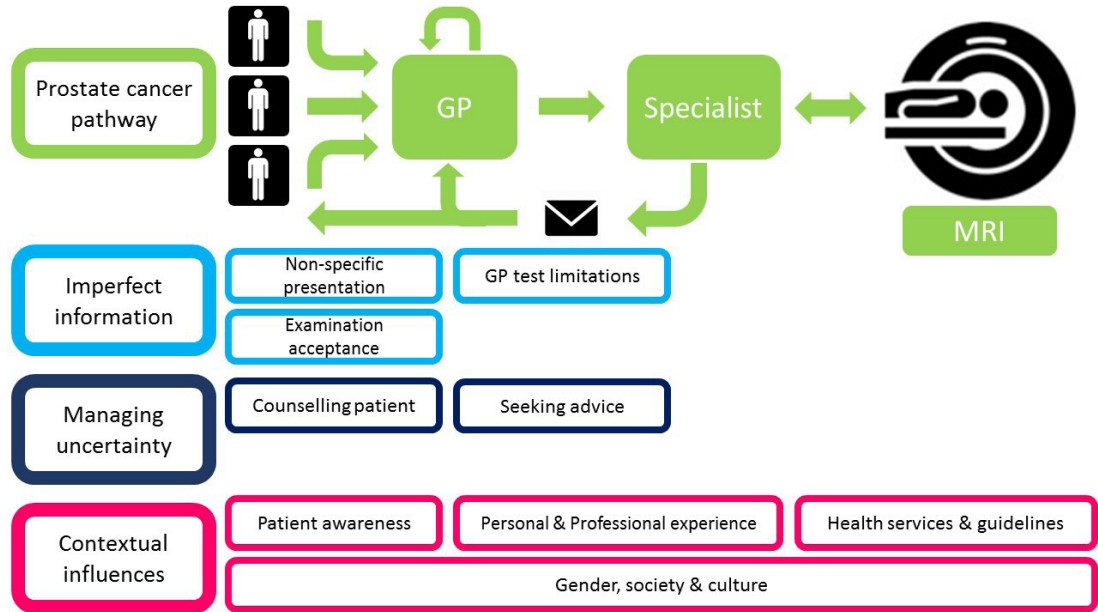

**Figure 4** Thematic diagram from GP participant interviews. GP, general practitioner.

about having *imperfect information* on which to base their clinical decisions.

A few GPs described a sense of inevitability about patients presenting with lower urinary tract symptoms at some point as they entered their later years (*Non-specific presentation*):

> It's a bit of a grey area so you're kind of waiting for patients to develop symptoms and come to see you. GP03 (Male, London, 31–40)

As described earlier, GPs experienced men refusing to have a prostate examination when prostate cancer is suspected (*Examination acceptance*). GPs reported different reasons for this, and perceived that patients may still be worried even if the prostate feels normal:

> I've had patients before who even will have got a high PSA decline, a rectal examination because they've previously had some, kind of, you know, traumatic experience or whatever. GP04 (Female, Devon, 41–50)

GPs from both regions did not hold back in sharing their opinions about the PSA blood test, and its usefulness (or lack thereof) in helping them make clinical decisions about which men to refer for further testing for possible prostate cancer. PSA appeared to hold poor face validity with GPs, and they expressed a hesitance in ordering the test (*GP test limitations*):

> I think if there's one test you could un-invent, I think PSA would be that… GP02 (Male, Devon, 31–40)
>
> So it's [PSA] quite a pain in the neck actually, to be honest… GP05 (Female, Devon, 41–50)
>
> Well, I don't like doing the PSA levels I suppose is one thing to say. GP07 (Female, London, 31–40)

GPs working in the NHS cannot currently order an MRI of the prostate; the request must come from a secondary or tertiary care clinician. London GPs were more likely to be positive about the concept of a prostate MRI:

> I think it will be a really useful idea. GP03 (Male, London, 31–40)
>
> Well, it's great, but it's not available to me. It's not something I decide on. GP05 (Female, Devon, 41–50)

### Managing uncertainty

GPs made efforts to share their diagnostic dilemma with patients where possible and consulted guidelines and their local urology specialists in managing uncertainty in their decisions about which men to refer to secondary care. Prior to referral, GPs tried to make their patients understand the limitations of the current diagnostic pathway (*counselling patient*):

> But I always would tell patients that it's not 100% and that both my examinations, whether it's a digital rectal or a PSA, are not 100% and it can be raised even without having cancer. GP03 (Male, London, 31–40)

While most GPs reported feeling satisfied with their local urology service (see *health service & guidance* below), some Devon GPs reported inconsistencies in the advice and management plans for their patients that came back from hospital specialists (*seeking advice*):

> I mean, we try to follow the guidelines but, as I say, we find mystifying as to the variation in the urology advice that comes back in terms of who to follow and who not to… GP04. (Female, Devon, 41–50)

## Contextual influences

A spectrum of broader influences had an effect on when patients chose to present to their GP with concerns about prostate cancer, and the consultation itself (*Gender, society & culture*). Some GPs noted a reticence of men to seek healthcare:

I think men don't… it's such a sweeping statement but men don't like coming to the doctor. GP07 (Female, London, 31–40)

Consistent with the patient interviews (*outside influences*), the GPs reported that it was often the wives and partners encouraging male patients to seek help and advice:

…the majority of men I see who mention prostate cancer it's because their wives have asked them to come and they're worried. GP07 (Female, London 31–40)

Cultural and ethnic norms relating to the patient and their partners also influenced the consultation and acceptance of prostate examination, which were more commonly reported by GPs working in London:

And over here I notice there are some patients of south Indian descent where, it's [DRE] almost like a taboo really. GP03 (Male, London, 31–40)

GPs in both regions were aware of the influence of news and media stories relating to prostate cancer that were encouraging patients with symptoms or concerns to see their GP and get tested:

…there was a lot in the media recently with prostate and testicular cancer, actually which is a good thing, because we had a… I had suddenly quite a few men coming in requesting the blood test. GP09 (Female, London, 41–50)

GPs felt that most patients were aware of prostate cancer and that tests were available for it. Awareness of MRI of the prostate was lower than for the PSA blood test (*Patient awareness*):

Lots of people are aware of the PSA. GP07 (Female, London, 31–40)

I think a few of them might have said, I've heard there's a new test around. I don't think anyone's come in and said, I'd like to have that MRI test. GP04 (Female, Devon, 41–50)

The decision-making of GPs was also affected by their own experiences in their personal and professional lives (*Personal & professional experience*). GPs demonstrated an awareness of how these experiences shaped their approach:

…my dad has prostate cancer that was picked up with a raised PSA. And my stepfather has prostate cancer which was picked up by a raised PSA. Both completely asymptomatic. So I think that also affects how you…

how you practice and you know, as clinicians we do take on our life experiences and we can't help but have that shape how… how we work. GP07 (Female, London, 31–40)

The health service context in which GPs practise was another significant influence on their approach to patients with possible prostate cancer (*Health services & guidelines*). They often rely on guidance from a number of sources, including national guidelines and local urology services:

I think we've got some, you know, very good local colleagues who offer good pragmatic advice and are very approachable. GP02 (Male, Devon, 31–40)

## DISCUSSION
### Principal findings

Patient and GP experiences of more traditional and 'one-stop' prostate cancer diagnostic pathways incorporating MRI showed some key similarities and differences. Communication was a key element in the experience of the prostate cancer diagnostic pathway for both patients and GPs. The communication between patients and healthcare teams significantly affected the patients' overall experience and their understanding of MRI results. GPs valued the ability to communicate with specialists to obtain pragmatic advice and guidance, particularly in the context of their hesitancy in relying on PSA test results. Family and personal experiences also shaped the awareness of both patients and GPs in relation to prostate cancer diagnosis.

Compared with patients attending a 'one-stop' clinic, patients following more traditional diagnostic pathways felt that longer waits for tests, more appointments to attend and increased travel requirements all impacted on their pathway experience. GPs faced challenges in dealing with uncertainty and the perceived limitations of symptoms, examination and tests available to them for diagnosing prostate cancer with confidence. GP awareness, understanding and access relating to MRI was limited in both regions.

### Relation to published literature

This is the first study that the authors are aware of to explore experiences of the modern prebiopsy MRI prostate cancer diagnostic pathway from the perspective of patients and GPs. Ruseckaite *et al*[18] interviewed 10 GPs from metropolitan Melbourne and a regional part of Victoria, Australia in 2015 regarding their perceptions of prostate cancer care. In line with the findings of this study, most men were willing to have a PSA blood test, and some GPs had to grapple with inconsistent guidance from specialist bodies. Evans *et al*[19] assessed men's experience of PSA testing in primary care in Wales in 2003–2004, and also found that social networks and media stories influenced patient demand for testing. In contrast to the views

of GPs in this study, the men in the study by Evans et al[19] felt decision-making about testing was doctor-centred rather than shared or patient-centred.

In contrast to the limited amount of published evidence on patient experience of cancer diagnostic pathways, there have been many more studies on patient perspectives of prostate cancer screening that identified some key themes consistent with the findings of this study.[20] James et al found 'social prompting' from family and friends to consult their doctor about prostate cancer testing is a prominent theme in prostate cancer screening studies, similar to the 'outside influences' subtheme that came from this research. Interestingly, patients in prostate cancer screening studies also describe the 'physiological and symptomatic obscurity' of prostate cancer, which the GPs in this study were acutely aware of.

'Communication' of the results of diagnostic testing and a new diagnosis of cancer was another key theme emerging from interviews in this study that has a wealth of published research, and quality of communication can impact on patient and clinician experiences of diagnostic pathways. A number of studies have found deficiencies in communication from clinicians to patients about prostate cancer diagnostic testing and the results of tests, similar to the experience of some patients interviewed.[21 22] Some patients had the opportunity to discuss test results and understand the implications of the findings when they had it, while others felt communication about their test results was largely bypassing them between the specialist and GP. Interventions for improving prostate cancer patient engagement and empowerment have previously been developed,[23] which may have a role in improving patient experience of modern prostate cancer diagnostic pathways.

### Strengths and weaknesses

This study recruited a diverse sample of patients undergoing prostate MRI in terms of age, geographical distribution and ethnicity. Recruitment of patients from a range of ethnic minority backgrounds in cancer research can be challenging,[24] so identifying and interviewing these patients as part of the study was key. Participants were recruited from two regions with contrasting prostate cancer diagnostic pathway designs. This enabled identification of key similarities and differences in the experiences of patients and GPs engaging with 'one stop' and more traditional pathways to help inform pathway design that could improve patient experience.

The influence of the researcher on data collection and analysis is important to consider in qualitative research. Participants were aware that SWDM was a clinician performing the study as part of a Cancer Research UK funded PhD, and that may have given some level of respectability and authority to the interviewer and the study. Some patients and GPs reported that men were less comfortable seeing a female GP about problems relating to the prostate, so having a male interviewer may have helped patient participants be more comfortable and

open in the interviews. GP participants may have been more comfortable in talking to a peer in these interviews; peer discussions are a common part of professional practice for GPs in the form of Balint groups[25] and annual appraisal by a fellow GP.[26]

Some GPs were reluctant to engage in any discussion about prostate MRI as they felt it was outside their current scope of practice and may have been focused on the more traditional (pre-MRI) prostate cancer pathway. MRI is a new test for prostate cancer and has only recently been integrated into diagnostic pathways. GPs are not currently able to request an MRI of the prostate, and access to MRI for other indications varies across the NHS. In this context, data gathered from GP participants were not as rich as the data collected from the patients and more limited insights were generated. A further potential limitation to the clinician insights gained from this study is that only GPs were recruited, and not other clinicians involved in the prostate cancer diagnostic pathway such as urologists or radiologists

### Implications for patients, clinicians and health service design

Men's experiences of the prostate cancer diagnostic pathway are influenced by the appointment burden they face to receive a diagnosis; the mode of communication used by GPs and specialists to communicate test results and requirements for travel to attend clinic appointments and tests. Significant challenges remain for GPs owing to the limitations of the current clinical signs and tests they rely on to identify possible prostate cancer cases. Men seemed broadly positive about MRI as a new test for prostate cancer, whereas GPs were equivocal owing to a lack of awareness and access. Improvements to patient experience of prostate cancer diagnostic pathways could be achievable through shorter time intervals to MRI, reduced outpatient appointment burden for patients and access to more accurate and reliable triage testing in primary care.

**Author affiliations**
[1]Institute of Health Research, University of Exeter, Exeter, UK
[2]Department of Public Health and Primary Care, University of Cambridge, Cambridge, UK
[3]Department of Psychology, University of Cambridge, Cambridge, UK
[4]Department of Behavioural Science and Health, University College London, London, UK
[5]Imperial Prostate, Imperial College London, London, UK
[6]Department of Urology, Royal Devon and Exeter NHS Foundation Trust, Exeter, UK
[7]Wolfson Institute of Population Health, Queen Mary University of London, London, UK

**Correction notice** The article was corrected since it was published online. The author affiliation #7 has been amended to Wolfson Institute of Population Health, Queen Mary University of London, London, UK.

**Acknowledgements** The authors would like to thank the participants for sharing their time and experiences for this study. We would also like to thank Pauline Sibley and Victor Mariano (Royal Devon & Exeter NHS Foundation Trust) for their assistance with patient recruitment.

**Contributors** SWDM, WH and FMW conceived the study. SWDM developed the research protocol with contributions from FMW, ASF and SA. HUA, JMG and DE-E were local investigators who supervised recruitment into the study. SWDM

undertook all interviews. SWDM, FMW, ASF and SA performed the analysis. SWDM drafted the first version of the manuscript. All authors contributed to the development of the manuscript, and approved the final submitted version. SWDM accepts full responsibility for the work and/or the conduct of the study, had access to the data, and controlled the decision to publish.

**Funding** SWDM is supported by the Can Test Collaborative, which is funded by CRUK (C8640/A23385). FMW and WH are codirectors of CanTest. HUA's research is supported by core funding from the UK's National Institute of Health Research (NIHR) Imperial Biomedical Research Centre (no award/grant number). HUA currently receives funding from the Wellcome Trust, Medical Research Council (UK), Cancer Research UK, Prostate Cancer UK, National Institute for Health Research (UK), The Urology Foundation, BMA Foundation, Imperial Health Charity, NIHR Imperial BRC, Sonacare, Trod Medical and Sophiris Biocorp for trials in prostate cancer. DE-E receives funding from the Urology Foundation, the BMA Foundation for Medical Research, Imperial Health Charity and the Royal College of Surgeons of England.

**Competing interests** HUA was a paid medical consultant for Sophiris Biocorp in the previous 3 years. The remaining authors have no conflicts of interest to declare.

**Patient and public involvement** Patients and/or the public were involved in the design, or conduct, or reporting or dissemination plans of this research. Refer to the Methods section for further details.

**Patient consent for publication** Not required.

**Ethics approval** This study involves human participants and ethics committee approval was received from the NHS HRA South-West Frenchay research ethics committee (REC reference 19/SW/0040). Participants gave informed consent to participate in the study before taking part.

**Provenance and peer review** Not commissioned; externally peer reviewed.

**Data availability statement** Data are available to bone fide researchers upon reasonable request. All data requests should be submitted in writing to the corresponding author.

**ORCID iDs**
Samuel William David Merriel http://orcid.org/0000-0003-2919-9087
Stephanie Archer http://orcid.org/0000-0003-1349-7178
Alice S Forster http://orcid.org/0000-0002-9933-7919
Hashim Uddin Ahmed http://orcid.org/0000-0003-1674-6723
Willie Hamilton http://orcid.org/0000-0003-1611-1373
Fiona M Walter http://orcid.org/0000-0002-7191-6476

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
