## [Reviewer comments · BMJ Open]

ARTICLE DETAILS

TITLE (PROVISIONAL)	Experiences of 'traditional' and 'one-stop' MRI-based prostate cancer diagnostic pathways in England: a qualitative study with patients and GPs
AUTHORS	Merriel, Samuel; Archer, Stephanie; Forster, Alice; Eldred-Evans, David; McGrath, John; Ahmed, Hashim; Hamilton, Willie; Walter, Fiona

VERSION 1 – REVIEW

REVIEWER	Pujadas Botey, Anna Alberta Health Services Board, Cancer Strategic Clinical Network
REVIEW RETURNED	04-Aug-2021

GENERAL COMMENTS	Thank you for the opportunity to review this paper. Overall, it is well written and deals with an important and well-researched topic: diagnostic pathways for patients with a potential prostate cancer diagnosis. However, it is not clear how the article contributes to the existing literature or practice, and the novelty of new findings is not clearly stated. Furthermore, the methods are not completely clear, and the discussion is limited. Some additional comments and considerations are listed below: - The authors need to better integrate this research to some of the existing research in the introduction and discussion sections. The introduction lacks an overview of the literature in the topic, and the discussion presents a limited explanation about how the results compare to earlier findings.- The objective of the study is somewhat vague. I wonder if using a theoretical framework identifying the different intervals from cancer suspicion to diagnosis or treatment could be helpful to add more precision. See, for example, the "Model of Pathways to Treatment" framework, in: Scott SE, Walter FM, Webster A, Sutton S, Emery J. The model of pathways to treatment: Conceptualization and integration with existing theory. Br J Health Psychol. 2013;18(1):45-65.- The justification and importance of the study is not clearly presented.- Details about how the contextual factors of the particular NHS Trusts and regions selected or particular factors related to the English health system in general may have contributed to the findings are not presented or discussed. Thus, implications of findings are uncertain.- The article is descriptive, yet with this qualitative work the authors do not present implications and outline recommendations for next
---

	steps –based on this data, what types of interventions could be proposed/developed, and what other factors need to be considered with intervention development? - The constructivist methodology should be presented as an approach that suggests that reality is socially constructed, and its adoption should be justified by its alignment with the objective of the study rather than the authors' assumption that participants "would construct their views based in their experiences...". In addition, the methodological approach is not something that the authors validate with data collected -as suggested in lines 31-37 of page 2-, but the approach used to access data and make sense of them. - It is not clear why the two particular NHS trusts were selected. The only explanation provided is that the two follow the two different pathway approaches included in the study. Since -I assume- there are many other NHS trusts following one or the other approach, further justification might be needed. - The methods section lacks important details. For example, how were patient participants contacted? How was the study promoted to local practices? How did interested GPs contact the team? How was eligibility of participants ensured? How was purposive sampling used to obtain diversity of participants? How were participants selected? Were interviews pilot-tested (how)? What were field notes used for and how? Had the research team an established relationship with participants prior to the start of the study? Did participants know anything about the researchers before participating in the study? Are there any researcher bias or assumptions that need to be disclosed? Why did some participants approached to participate declined participation (and how was that 'solved')? Who analyzed the interviews that were analyzed after the selected "early interviews"? How were experiences of participants "compared and contrasted" (is this the right wording)? What were particular methods used to ensure trustworthiness? Who did the coding (if more than one investigator, how were disagreements solved)? Were interview transcripts returned to participants for comments/feedback? The authors explain that a study summary report was sent to all participants, but the purpose of that action is not explained; was it to validate results or seek comments/feedback? The authors mention that recruitment ceased when no new themes emerged in the analysis (i.e., data saturation), but it is not clear how recruitment, purposeful selection of participants, and interview analysis were organized and coordinated. - The statement in the results section "SM conducted all the interviews for this study", their occupation and gender belong to the methods section. - In the results section, it is indicated that patients could "choose to have their wives present and involved in the interview". I wonder if the authors could reconsider the wording to be considerate of the plausible situation where both the patient and their wife (or partner), or the wife (or partner) alone chose to participate. - In some instances, the wording is not clear. For example, what does "patient participants were mostly interviewed face-to-face in their own home" mean? The statement in the strengths and limitations point section "limited knowledge of prostate MRI curtailed interviews with some GP participants" might need rewording.
--	---

	- The results are very succinctly presented. I suggest that the space assigned to the results in the body of the manuscript is used to explain the emerging themes in more depth, and –if needed- the quotations are presented in a Table. - Most of the results about patients and GPs' perspectives are applicable to general experiences of patients and GPs potentially diagnosed with cancer (no matter what kind of cancer, diagnostic pathway followed, or particular testing used) –cancer beliefs and expectations, communication/information, and logistics/complications with testing are widely reported in the literature. I wonder if the authors want to reconsider what they are presenting in this section to better discuss the experiences of patients and GPs in relation to the inclusion of pre-biopsy MRI in different pathways, which seems to be the focus of the study as presented in the introduction. In addition, these results should be discussed in the discussion section, in the context of published similar studies (now in 2nd and last paragraphs, page 21). This mismatch between the objective of the study and the results (and discussion) might be related to the comment above about the lack of clarity when stating the objective of the study. - I am not very familiar with the English health system, and the particular case of prostate cancer, but I am surprised to see that the interaction between primary and specialist care did not come up in the study results (and discussion). In the cancer diagnosis literature, it has been widely reported that GPs struggle to effectively communicate with specialists for advice and referral, and to get cancer diagnosis testing done in timely and orderly fashion. - The strengths and weaknesses piece of the discussion section needs to be rewritten to clearly present the strengths and weaknesses of the study. The 1st paragraph of the current draft rephrases the objective and methods of the study, the 2nd paragraph presents some additional information related to the methods, and the 3rd paragraph presents some additional information related to the introduction. - Given the nature of the study (qualitative research) careful wording should be used to better express the idea of comparison between groups of patients (e.g., sentence in discussion section: “compared to patients attending a ‘one-stop’ clinic, patients following more traditional diagnostic pathways felt that longer waits for tests, more appointments to attend, and increased travel requirements all impacted on their pathway experience”; sentence in strengths and limitations point section: “Patient experiences of two very different prostate cancer diagnostic pathways compared and contrasted”).
--	---

REVIEWER	Cuesta-Briand, Beatriz The University of Western Australia, Rural Clinical School of WA
REVIEW RETURNED	07-Aug-2021

GENERAL COMMENTS	Thank you for your submission. The manuscript presents valuable data that has the potential to make a valuable contribution to the body of knowledge on prostate cancer diagnostic pathways. However, the paper presents several major issues that ought to be addressed. Background: this section provides a brief overview of the topic but fails to position it within the existing literature. In particular, I would
---

	have liked to see a reference to the literature on patients' (and GPs') experiences of prostate cancer diagnosis, in particular qualitative data. Although the figures are informative, using a similar format for both would have allowed for an easier comparison between the two pathway models, especially regarding time frames. Methods It is unclear how the 'constructivist approach' informed the data analysis process. Results Whilst the authors present the results in a complex diagrammatic form in Figures 3 & 4, this complexity is not translated to the actual narrative presentation of results. There is little evidence of thematic analysis beyond the identification of categories of topics ('cancer beliefs', 'communication'), and the reader does not get a real sense of the level of agreement between participants, contrasting opinions and experiences, etc. Also, the differences between the experiences of the two groups of patients are only superficially explored under the heading 'pathway experience'. The results section would benefit from a better integration of the experiences of GPs and patients, and from a more in-depth exploration of the differences between users' experiences of the two pathway models (both GPs and patients). Discussion In its current format, the discussion section is too fragmented and lacks depth. More emphasis should be placed on how the results fit within the existing literature, and the impact of the socio-cultural context (as cited by the authors in the methods section). The last section on 'implications' draws conclusions that are too broad and overreach beyond the evidence presented.
--	--

VERSION 1 – AUTHOR RESPONSE

Reviewer: 1

Dr. Anna Pujadas Botey, Alberta Health Services Board Comments to the Author:

Thank you for the opportunity to review this paper. Overall, it is well written and deals with an important and well-researched topic: diagnostic pathways for patients with a potential prostate cancer diagnosis. However, it is not clear how the article contributes to the existing literature or practice, and the novelty of new findings is not clearly stated. Furthermore, the methods are not completely clear, and the discussion is limited.

Thank you for taking the time to review this manuscript, and for your helpful comments. We have tried to address the reviewers' comments where possible, in the context of this manuscript already slightly exceeding the journal's word count limit.

Some additional comments and considerations are listed below:

- The authors need to better integrate this research to some of the existing research in the introduction and discussion sections. The introduction lacks an overview of the literature in the topic, and the discussion presents a limited explanation about how the results compare to earlier findings.

Thank you for highlighting the need to better explain the place of this research in the area of patient experience. Please see the revised Introduction, particularly the opening paragraph, which addresses this comment.

- The objective of the study is somewhat vague. I wonder if using a theoretical framework identifying the different intervals from cancer suspicion to diagnosis or treatment could be helpful to add more precision. See, for example, the "Model of Pathways to Treatment" framework, in: Scott SE, Walter FM, Webster A, Sutton S, Emery J. The model of pathways to treatment: Conceptualization and integration with existing theory. Br J Health Psychol. 2013;18(1):45-65.

The Model of Pathways to Treatment is an important seminal paper in understanding diagnostic pathways. In order to make it clearer to the reader where this study sits within cancer diagnostic pathways we have added reference to this framework.

- The justification and importance of the study is not clearly presented.

The Introduction has been revised to make the justification for the study clearer, in particular the third paragraph.

- Details about how the contextual factors of the particular NHS Trusts and regions selected or particular factors related to the English health system in general may have contributed to the findings are not presented or discussed. Thus, implications of findings are uncertain.

The optimal way of implementing prostate MRI into current clinical pathways is currently unclear, both from a clinical effectiveness perspective and in terms of patient and clinician experience and acceptability. This is discussed in the Introduction. These two Trusts were chosen for patient recruitment as they have implemented prostate MRI in very different ways (see Figures 1 and 2 in the Methods Section), and presented an opportunity to compare patient and clinician experiences between the two pathways. This justification has been made clearer in the 'Recruitment' sub-section of the Methods.

Further discussion about the potential implications for prostate cancer diagnostic pathway design has been added to the 'Implications for clinicians and health service design' section in the Discussion.

- The article is descriptive, yet with this qualitative work the authors do not present implications and outline recommendations for next steps –based on this data, what types of interventions could be proposed/developed, and what other factors need to be considered with intervention development?

The purpose of this research is not to inform the development of new interventions for prostate cancer diagnosis. Prostate MRI is a new test for prostate cancer that has strong evidence for diagnostic accuracy, but other key areas for optimal implementation of prostate MRI (including patient and clinician experience and acceptability, and diagnostic pathway design) are less well understood and explored. This interview study adds to our understanding of patient and clinician experiences of two very differently designed prostate cancer diagnostic pathways, which will help inform the integration of prostate MRI into diagnostic pathways across the NHS and internationally. This has been made clearer in the 'Implications for clinicians and health service design' sections of the Discussion.

- The constructivist methodology should be presented as an approach that suggests that reality is socially constructed, and its adoption should be justified by its alignment with the objective of the study rather than the authors' assumption that participants "would construct their views based in their experiences...". In addition, the methodological approach is not something that the authors validate with data collected -as suggested in lines 31-37 of page 2-, but the approach used to access data and make sense of them.

Thank you for highlighting this important distinction. The first paragraph of the methods has been amended to make this clearer, and the strengths and weaknesses section rewritten to correct the description of how the methodological approach helped make sense of the data.

- It is not clear why the two particular NHS trusts were selected. The only explanation provided is that the two follow the two different pathway approaches included in the study. Since -I assume- there are many other NHS trusts following one or the other approach, further justification might be needed.

As discussed earlier, these two Trusts were chosen for patient recruitment as they have implemented prostate MRI in very different ways (see Figures 1 and 2 in the Methods Section), and presented an opportunity to compare patient and clinician experiences between the two pathways. This justification has been made clearer in the 'Recruitment' sub-section of the Methods.

- The methods section lacks important details. For example, how were patient participants contacted? How was the study promoted to local practices? How did interested GPs contact the team? How was eligibility of participants ensured? How was purposive sampling used to obtain diversity of participants? How were participants selected? Were interviews pilot-tested (how)? What were field notes used for and how? Had the research team an established relationship with participants prior to the start of the study? Did participants know anything about the researchers before participating in the study? Are there any researcher bias or assumptions that need to be disclosed? Why did some participants approached to participate decline participation (and how was that 'solved')? Who analyzed the interviews that were analyzed after the selected "early interviews"? How were experiences of participants "compared and contrasted" (is this the right wording)? What were particular methods used to ensure trustworthiness? Who did the coding (if more than one investigator, how were disagreements solved)? Were interview transcripts returned to participants for comments/feedback? The authors explain that a study summary report was sent to all participants, but the purpose of that action is not explained; was it to validate results or seek comments/feedback? The authors mention that recruitment ceased when no new themes emerged in the analysis (i.e., data saturation), but it is not clear how recruitment, purposeful selection of participants, and interview analysis were organized and coordinated.

We agree with the reviewer that all of these elements are important, and they are included in the study protocol. Unfortunately due to the word count limit of the journal, we cannot include all this information in the main manuscript. The approved study protocol has been included as a supplementary file, to make the requested information on the methods available.

- The statement in the results section "SM conducted all the interviews for this study", their occupation and gender belong to the methods section.

The statement above was in the Discussion section, but has been removed in this revised version. It is also mentioned in the 'data collection' sub-section of the methods, where the occupation and gender of the interviewer is indicated. No mention of who conducted the interviews features in the Results section.

- In the results section, it is indicated that patients could "choose to have their wives present and involved in the interview". I wonder if the authors could reconsider the wording to be considerate of the plausible situation where both the patient and their wife (or partner), or the wife (or partner) alone chose to participate.

Thank you for the suggestion. This sentence has been reworded to be more balanced and objective.

- In some instances, the wording is not clear. For example, what does "patient participants were mostly interviewed face-to-face in their own home" mean? The statement in the strengths and limitations point section "limited knowledge of prostate MRI curtailed interviews with some GP participants" might need rewording.

Thank you for highlighting these sentences, which have been reworded for greater clarity.

- The results are very succinctly presented. I suggest that the space assigned to the results in the body of the manuscript is used to explain the emerging themes in more depth, and –if needed- the quotations are presented in a Table.

Thank you for your suggestions to improve the results section. Data obtained from the interviews that is less relevant to the study objective have been removed, allowing for more in-depth analysis and explanation of the basis for the emerging themes.

- Most of the results about patients and GPs' perspectives are applicable to general experiences of patients and GPs potentially diagnosed with cancer (no matter what kind of cancer, diagnostic pathway followed, or particular testing used) –cancer beliefs and expectations, communication/information, and logistics/complications with testing are widely reported in the literature. I wonder if the authors want to reconsider what they are presenting in this section to better discuss the experiences of patients and GPs in relation to the inclusion of pre-biopsy MRI in different pathways, which seems to be the focus of the study as presented in the introduction. In addition,

these results should be discussed in the discussion section, in the context of published similar studies (now in 2nd and last paragraphs, page 21). This mismatch between the objective of the study and the results (and discussion) might be related to the comment above about the lack of clarity when stating the objective of the study.

Thank you for your suggestions about better alignment between the Introduction, study aims, and Results presented. As discussed earlier, the Introduction and statement of aims has been revised for greater clarity for the reader. Pre-biopsy MRI is one test within a diagnostic pathway, as shown in figures 3 and 4 and the Model of Pathways to Treatment by Scott *et al.* We were conscious of this in the data collection and analysis, and this is borne out in the data which, as you rightly highlight, covers a broader range of themes than just the experience of MRI. Some of the themes that emerged could hold relevance for other cancers, but equally there are some unique elements to prostate cancer pathways including the appointment burden faced for patients in the more traditional pathway and GP access to MRI. Greater emphasis has been placed on the findings in relation to experience of MRI within the pathway to align with the remainder of the manuscript.

- I am not very familiar with the English health system, and the particular case of prostate cancer, but I am surprised to see that the interaction between primary and specialist care did not come up in the study results (and discussion). In the cancer diagnosis literature, it has been widely reported that GPs struggle to effectively communicate with specialists for advice and referral, and to get cancer diagnosis testing done in timely and orderly fashion.

The interaction between primary and secondary care specialists did come up in the interviews and does feature in the Results section, in particular about inconsistencies in advice from local specialists adding to the uncertainty around making a diagnosis of prostate cancer. Please see the Managing Uncertainty theme from the GP interviews on page 17-18.

- The strengths and weaknesses piece of the discussion section needs to be rewritten to clearly present the strengths and weaknesses of the study. The 1st paragraph of the current draft rephrases the objective and methods of the study, the 2nd paragraph presents some additional information related to the methods, and the 3rd paragraph presents some additional information related to the introduction.

Thank you for your feedback on this section. The 'Strengths and weaknesses' have been revised.

- Given the nature of the study (qualitative research) careful wording should be used to better express the idea of comparison between groups of patients (e.g., sentence in discussion section: "compared to patients attending a 'one-stop' clinic, patients following more traditional diagnostic pathways felt that longer waits for tests, more appointments to attend, and increased travel requirements all impacted on their pathway experience"; sentence in strengths and limitations point section: "Patient experiences of two very different prostate cancer diagnostic pathways compared and contrasted").

Thank you for your suggestion. Wording to convey the use of comparison between patients and GPs' experience of different prostate cancer diagnostic pathways has been revised.

Reviewer: 2

Dr. Beatriz Cuesta-Briand, The University of Western Australia Comments to the Author:

Dear Authors

Thank you for your submission. The manuscript presents valuable data that has the potential to make a valuable contribution to the body of knowledge on prostate cancer diagnostic pathways.

Thank for taking the time to review this manuscript, and for your helpful comments.

However, the paper presents several major issues that ought to be addressed.

Background: this section provides a brief overview of the topic but fails to position it within the existing literature. In particular, I would have liked to see a reference to the literature on patients' (and GPs') experiences of prostate cancer diagnosis, in particular qualitative data.

Thank you for this suggestion. The Introduction has been revised to include literature on patient and GP experiences of prostate cancer diagnosis.

Although the figures are informative, using a similar format for both would have allowed for an easier comparison between the two pathway models, especially regarding time frames.

Methods

It is unclear how the 'constructivist approach' informed the data analysis process.

Thank you for your feedback. The first paragraph of the Methods section has been amended to make this clearer.

Results

Whilst the authors present the results in a complex diagrammatic form in Figures 3 & 4, this complexity is not translated to the actual narrative presentation of results. There is little evidence of thematic analysis beyond the identification of categories of topics ('cancer beliefs', 'communication'), and the reader does not get a real sense of the level of agreement between participants, contrasting opinions and experiences, etc. Also, the differences between the experiences of the two groups of patients are only superficially explored under the heading 'pathway experience'. The results section would benefit from a better integration of the experiences of GPs and patients, and from a more in-depth exploration of the differences between users' experiences of the two pathway models (both GPs and patients).

Thank you for your suggestions to improve the results section. Data obtained from the interviews that is less relevant to the study objective have been removed, allowing for more in-depth analysis and explanation of the basis for the emerging themes and differences in experiences between the two pathways.

Discussion

In its current format, the discussion section is too fragmented and lacks depth. More emphasis should be placed on how the results fit within the existing literature, and the impact of the socio-cultural context (as cited by the authors in the methods section). The last section on 'implications' draws conclusions that are too broad and overreach beyond the evidence presented.

Thank you for your feedback. The Discussion section has been revised to be more consistent with the findings of the study and more emphasis on placing the findings within existing literature on patient and GP experiences of cancer diagnosis pathways.

VERSION 2 – REVIEW

REVIEWER	Pujadas Botey, Anna Alberta Health Services Board, Cancer Strategic Clinical Network
REVIEW RETURNED	19-Nov-2021
GENERAL COMMENTS	Thank you for the opportunity to review this paper again. My initial feedback and concerns have been mostly addressed with the exception of three points that I think could be considered to strengthen the manuscript. They are: - Discussion of the importance and justification of the study. The objective is now clearly presented and some information framing this objective has been added. However, I am still missing a statement that highlights the benefit of “eliciting the experience of patients and GPs following two prostate cancer diagnostic pathways that incorporate pre-biopsy MRI in different ways”. In the authors’ previous response to the reviewers’ comments, they stated that this information will help inform the integration of prostate MRI into diagnostic pathways across the NHS and internationally. I think this

	is an important contribution that should be used to (at least partially) justify the study, and should be further elaborated in the discussion or conclusion of the study (i.e., how does this study inform that? what are next steps from this study to get into this integration?) - I feel the methods section still lacks important details. Adding the study protocol as a supplemental material is a great idea, and helps with some additional details, but a lot of critical information about the methods remains unknown. I suggest that the authors go back to the COREQ checklist, and make sure all criteria for reporting qualitative research are actually addressed in the manuscript (or supplemental material). - I do not think that the strengths and weaknesses section is presenting the strengths and weaknesses of the study. The 1st paragraph presents a justification of recruitment, 2nd paragraph presents some additional information related to the methods, and the 3rd paragraph presents some additional information related to the introduction (with the exception of the last sentence). This section should present the strengths of the study, and discuss its flaws and steps taken to mitigate their effect.
--	--

REVIEWER	Cuesta-Briand, Beatriz The University of Western Australia, Rural Clinical School of WA
REVIEW RETURNED	17-Nov-2021

GENERAL COMMENTS	Thank you for submitting your revised manuscript. I acknowledge that some of the reviewers' queries have been addressed, however, the manuscript still has some flaws that ought to be addressed adequately. The major problem with the revised manuscript is that it still fails to adequately discuss the results in the context of the existing evidence. I note that some additional references have been added to the discussion, however, the authors fail to demonstrate their understanding of the literature on the experience of cancer diagnostic pathways. There is a wealth of qualitative evidence exploring patients' experience of prostate cancer screening diagnosis that the authors could have referenced (see systematic review: James LJ, Wong G, Craig JC, Hanson CS, Ju A, Howard K, et al. (2017) Men's perspectives of prostate cancer screening: A systematic review of qualitative studies. PLoS ONE 12(11): e0188258. https://doi.org/10.1371/journal.pone.0188258). Similarly, much has been written about patient/doctor communication in cancer diagnosis and care, and yet this issue is not fully addressed - the discussion section includes a few mentions of communication issues, but the findings are not discussed in relation to the literature on this topic, despite the fact that 'communication' is one of the main themes identified in the results section. Overall, the manuscript does not provide an adequate overview of the existing evidence on this topic in the introduction and then fails to adequately discuss the findings in the context of this evidence in the discussion section. In my opinion, the manuscript is not suitable for publication in its current form and requires major revision.
--

VERSION 2 – AUTHOR RESPONSE

Reviewer: 1

Dr. Anna Pujadas Botey, Alberta Health Services Board Comments to the Author:
Thank you for the opportunity to review this paper again.

Thank you for taking the time to review our revised manuscript.

My initial feedback and concerns have been mostly addressed with the exception of three points that I think could be considered to strengthen the manuscript. They are:

- Discussion of the importance and justification of the study. The objective is now clearly presented and some information framing this objective has been added. However, I am still missing a statement that highlights the benefit of “eliciting the experience of patients and GPs following two prostate cancer diagnostic pathways that incorporate pre-biopsy MRI in different ways”. In the authors’ previous response to the reviewers’ comments, they stated that this information will help inform the integration of prostate MRI into diagnostic pathways across the NHS and internationally. I think this is an important contribution that should be used to (at least partially) justify the study, and should be further elaborated in the discussion or conclusion of the study (i.e., how does this study inform that? what are next steps from this study to get into this integration?)

Thank you for the suggestion. This is touched upon in the opening paragraph of the Introduction. However, we have integrated further discussion about the value of this study in informing the design of prostate cancer diagnostic pathways into the Introduction and Discussion sections.

- I feel the methods section still lacks important details. Adding the study protocol as a supplemental material is a great idea, and helps with some additional details, but a lot of critical information about the methods remains unknown. I suggest that the authors go back to the COREQ checklist, and make sure all criteria for reporting qualitative research are actually addressed in the manuscript (or supplemental material).

We appreciate the reviewer’s desire for additional information about the study methods, all of which is now included in the manuscript and supplementary material. We have revisited the COREQ checklist and made further additions to the Methods section of the manuscript where possible.

- I do not think that the strengths and weaknesses section is presenting the strengths and weaknesses of the study. The 1st paragraph presents a justification of recruitment, 2nd paragraph presents some additional information related to the methods, and the 3rd paragraph presents some additional information related to the introduction (with the exception of the last sentence). This section should present the strengths of the study, and discuss its flaws and steps taken to mitigate their effect.

Thank you for your suggestion. The strengths and limitations have been revised further.

Reviewer: 2

Dr. Beatriz Cuesta-Briand, The University of Western Australia Comments to the Author:
Dear authors

Thank you for submitting your revised manuscript. I acknowledge that some of the reviewers' queries have been addressed, however, the manuscript still has some flaws that ought to be addressed adequately.

Thank you for taking the time to review the revised manuscript.

The major problem with the revised manuscript is that it still fails to adequately discuss the results in the context of the existing evidence. I note that some additional references have been added to the discussion, however, the authors fail to demonstrate their understanding of the literature on the experience of cancer diagnostic pathways. There is a wealth of qualitative evidence exploring patients' experience of prostate cancer screening diagnosis that the authors could have referenced (see systematic review: James LJ, Wong G, Craig JC, Hanson CS, Ju A, Howard K, et al. (2017)

Men's perspectives of prostate cancer screening: A systematic review of qualitative studies. PLoS ONE 12(11): e0188258. <https://eur03.safelinks.protection.outlook.com/?url=https%3A%2F%2Fdoi.org%2F10.1371%2Fjournal.pone.0188258&data=04%7C01%7CS.W.D.Merriel%40exeter.ac.uk%7Cc916aca21f8a4cde2c0a08d9d13e95f7%7C912a5d77fb984eeeaf321334d8f04a53%7C0%7C0%7C637770889000362557%7CUnknown%7CTWFpbGZsb3d8eyJWljojMC4wLjAwMDAiLCJQIjoiV2luMzliLCJBTiI6Ikh1aWwiLCJXVCi6Mn0%3D%7C3000&sd=0&data=XHV8b3Uo4c0QITFvSKCC0BGtcb6xV8Jj%2BS70sTGunHI%3D&reserved=0>).

Thank you to the reviewer for the suggestion for strengthening this manuscript further. An important distinction needs to be made between prostate cancer screening and diagnostic pathways following symptomatic presentation to a GP or primary care clinician. Screening programmes are delivered to healthy, asymptomatic adults in the general population to detect cancer at an early stage before symptoms appear. Prostate cancer screening is not recommended in the UK and this study was not focused on screening. Rather, this study sought to understand the experience of patients and GPs after a patient has presented to their GP concerned about their risk of prostate cancer or with symptoms that might relate to an undiagnosed prostate cancer, and the GP has referred the patient urgently for MRI and other diagnostic tests in the pathway. This is critical as there are key differences in cancer risk, symptom concern, and diagnostic approaches for asymptomatic individuals in the community compared to symptomatic patients presenting to healthcare services for further assessment, and patient experience may well differ as a result.

As the reviewer rightly points out, there is an abundance of studies on patient experience of prostate cancer **screening** programmes, even though very few countries actually have a formal prostate cancer screening programme. However, the vast majority of all cancer patients in the UK (including prostate cancer patients) are diagnosed following referral after **symptomatic** presentation in primary care. An additional limitation of existing studies of patient experience of prostate cancer screening is that almost all will be in the era before MRI was used as a diagnostic test; a gap which this study seeks to fill. Some of the lessons from studies of patient experience of prostate cancer screening may have some relevance to patient experience of symptomatic diagnosis and MRI-based diagnostic pathways. The authors have undertaken a further review of existing literature on patient experience of prostate cancer diagnostic pathways, critically assessed whether the findings are relevant to this study, and amended the Discussion to incorporate and contrast those findings that are potentially relevant.

Similarly, much has been written about patient/doctor communication in cancer diagnosis and care, and yet this issue is not fully addressed - the discussion section includes a few mentions of communication issues, but the findings are not discussed in relation to the literature on this topic, despite the fact that 'communication' is one of the main themes identified in the results section.

Thanks to the reviewer for the suggestion. The authors have further revised the Discussion section of the manuscript to compare the study findings around communication to relevant literature.

Overall, the manuscript does not provide an adequate overview of the existing evidence on this topic in the introduction and then fails to adequately discuss the findings in the context of this evidence in the discussion section. In my opinion, the manuscript is not suitable for publication in its current form and requires major revision.

Thank you for taking the time to review this revised manuscript. The authors hope all suggested revisions have been adequately addressed.

VERSION 3 – REVIEW

REVIEWER	Cuesta-Briand, Beatriz The University of Western Australia, Rural Clinical School of WA
REVIEW RETURNED	07-Mar-2022

GENERAL COMMENTS	Thank you for this opportunity to review the revised version of the manuscript. A few comments, which I hope will be helpful: Results: 1) Table 1: Spell out 'BME', as it is not an acronym that readers outside of the UK will be familiar with. 2) I would have liked to see a more in-depth discussion of results relating to GPs' views on PSA, supported with more quotes, especially given that the authors state that 'GPs did not hold back in sharing their opinions'. Discussion: 1) I would like to see a more in-depth discussion of the contrasting views of GPs and patients. This is potentially one of the strengths of the study, but it is not fully explored in the discussion. 2) Further, the weaving of the relevant literature into the discussion currently feels somewhat clunky and could be further refined (especially the section 'relation to published literature'). This would improve legibility and would allow the authors to focus on the main take-home messages. 3) The meaning of this sentence is unclear: 'This study found patients valued the opportunity to discuss test results and understand the implications of the findings when they had it, whilst others felt communication about their test results was largely bypassing them between the specialist and GP.' Overall, I would encourage the authors to reflect on what the main findings are (reflecting on how patients' views contrast (or not) with GPs' views) and then discuss how these findings fit within the existing literature. In my opinion, the current structure of the discussion detracts from legibility and clarity. Thank you for giving me the opportunity to review your revised manuscripts.
---

VERSION 3 – AUTHOR RESPONSE

Thank you for your further comments on our manuscript.

A few comments, which I hope will be helpful:

Results:

1) Table 1: Spell out 'BME', as it is not an acronym that readers outside of the UK will be familiar with.

Thank you for highlighting this. 'BME' has been replaced with 'non-white'

2) I would have liked to see a more in-depth discussion of results relating to GPs' views on PSA, supported with more quotes, especially given that the authors state that 'GPs did not hold back in sharing their opinions'.

We agree this is an important finding. Additional interview data has been added as well as further discussion in the Results and Discussion sections.

Discussion:

1) I would like to see a more in-depth discussion of the contrasting views of GPs and patients. This is potentially one of the strengths of the study, but it is not fully explored in the discussion.

Thank you for the suggestion. The key findings section of the Discussion has been expanded to further explore similarities and differences between the experience of patients and GPs.

2) Further, the weaving of the relevant literature into the discussion currently feels somewhat clunky and could be further refined (especially the section 'relation to published literature'). This would improve legibility and would allow the authors to focus on the main take-home messages.

Thank you for the suggestion. The weaving of relevant literature and the presentation of the findings from these studies has been refined to improve legibility and flow.

3) The meaning of this sentence is unclear: 'This study found patients valued the opportunity to discuss test results and understand the implications of the findings when they had it, whilst others felt communication about their test results was largely bypassing them between the specialist and GP.'

Thank you for highlighting this sentence. The start of the sentence has been amended to make it clearer how this finding relates to the references quoted in the previous sentence and the overall focus of the paragraph on communication of test results.

Overall, I would encourage the authors to reflect on what the main findings are (reflecting on how patients' views contrast (or not) with GPs' views) and then discuss how these findings fit within the existing literature. In my opinion, the current structure of the discussion detracts from legibility and clarity.

Thank you for the suggestion. In addition to the changes outline above, the order of the Discussion has been altered to improve clarity for the reader.

Thank you for giving me the opportunity to review your revised manuscripts.